# Dupilumab for the Treatment of Atopic Dermatitis in an Austrian Cohort-Real-Life Data Shows Rosacea-Like Folliculitis

**DOI:** 10.3390/jcm9041241

**Published:** 2020-04-24

**Authors:** Tamara Quint, Patrick M. Brunner, Christoph Sinz, Irene Steiner, Robin Ristl, Kornelia Vigl, Susanne Kimeswenger, Katharina Neubauer, Detlev Pirkhammer, Martin Zikeli, Wolfram Hoetzenecker, Norbert Reider, Christine Bangert

**Affiliations:** 1Department of Dermatology, Medical University of Vienna, 1090 Vienna, Austria; tamara.quint@meduniwien.ac.at (T.Q.); patrick.brunner@meduniwien.ac.at (P.M.B.); christoph.sinz@meduniwien.ac.at (C.S.); 2Institute of Medical Statistics, Medical University Vienna, 1090 Vienna, Austria; irene.steiner@meduniwien.ac.at (I.S.); robin.ristl@meduniwien.ac.at (R.R.); 3Department of Dermatology, Rudolfstiftung City Hospital, 1030 Vienna, Austria; kornelia.vigl@wienkav.at (K.V.); detlev.pirkhammer@wienkav.at (D.P.); 4Department of Dermatology, Kepler University Hospital, 4020 Linz, Austria; susanne.kimeswenger@gmail.com (S.K.); wolfram.hoetzenecker@kepleruniklinikum.at (W.H.); 5Department of Soft Matter Physics, Institute for Experimental Physics, Johannes Kepler University, 4020 Linz, Austria; 6Department of Dermatology, Medical University of Innsbruck, 6020 Innsbruck, Austria; katharina.neubauer@auva.at (K.N.); norbert.reider@i-med.ac.at (N.R.); 7Department of Dermatology, Wiener Neustadt Public Hospital, 2700 Wiener Neustadt, Austria; martin.zikeli@wienerneustadt.lknoe.at

**Keywords:** Dupilumab, atopic dermatitis, biologicals, conjunctivitis, rosacea-like folliculitis

## Abstract

Dupilumab is the first biological treatment approved for moderate-to-severe atopic dermatitis (AD). Efficacy and safety have been demonstrated in clinical trials, but real-life data is still limited. The objective of this study was to retrospectively evaluate Dupilumab treatment in AD patients in a real-life clinical setting. Effectiveness and safety outcomes were collected at baseline and after 2, 6, 10, 24, 39, and 52 weeks by using clinical scores for disease activity, as well as serological markers. Ninety-four patients from five dermatological hospitals were included. After 24 weeks of treatment, the median Investigator Global Assessment (IGA) and Eczema Area and Severity Index (EASI) showed a significant reduction compared to baseline (3.9 ± 0.7 vs. 1.4 ± 0.8 and 26.5 ± 12.5 vs. 6.4 ± 6.5). Interestingly, we observed rosacea-like folliculitis as an unexpected side effect in 6.4% of patients. Dupilumab proves to be an effective and well-tolerated treatment under real-life conditions. The occurrence of rosacea-like folliculitis warrants further mechanistic investigation.

## 1. Introduction

Atopic dermatitis (AD) is a chronic inflammatory skin disorder affecting up to 10% of the adult population and approximately 25% of children worldwide [1,2]. An aberrant innate immune response, environmental factors, and genetic susceptibility (such as loss-of-function mutations in the filaggrin gene) lead to epidermal barrier impairment and progressive formation of a strong T helper type 2 (Th2) immune response [3,4]. Approximately 20% of patients have moderate-to-severe symptoms of the disease characterized by intense itch and inflammatory eczematous skin lesions resulting in reduced quality of life [5]. The mainstay of treatment consists of emollients, topical corticosteroids (TCS), and various systemic immunosuppressive drugs, but these are not effective for all patients and can have substantial side effects [6,7,8].

Dupilumab, a fully human monoclonal antibody directed against the cytokine receptor component interleukin-4 receptor alpha(IL-4Rα), targets the pathogenic Th2 pathway by blocking effects of both interleukin (IL)-4 and IL-13. It is the first biologic treatment for AD approved by the food and drug administration (FDA) and European Medicines Agency (EMA) in 2017 based on clinical trials that have successfully proven its clinical safety and effectiveness [9,10,11]. Due to a strict patient selection policy in clinical trials, these data are not always reproducible in daily clinical practice, and real-world data may differ in effectiveness and safety [12]. Here, we present data of effectiveness and side effects of Dupilumab treatment in a real-life Austrian patient population.

## 2. Experimental Section

### 2.1. Study Design and Patients Included

This retrospective study evaluated data from AD patients treated at five independent dermatological centers in Austria, who started Dupilumab treatment between February 2018 and March 2019. All patients included were >18 years old, diagnosed with moderate-to-severe AD, according to the revised Hanifin and Rajka criteria, and received Dupilumab for inadequately controlled AD (Investigator Global Assessment Score 3 or 4) and inefficacy of previous systemic and topical treatments (Table 1) [13]. Concomitant topical treatment with either corticosteroids (TCS) and/or calcineurin inhibitors (TCI) was continued according to the investigators’ discretion during flares and was not specifically monitored. Patients were further asked to maintain their daily skin care using moisturizers. For previous unsuccessful systemic therapies, a washout period was applied according to the half time of the respective medication. Therefore, none of these patients received concomitant immunosuppressive systemic therapy or phototherapy. Dupilumab was prescribed according to the approved dosage of initially 600 mg followed by 300 mg every other week, applied subcutaneously. The study was approved by the local ethics committee (EK 1692/2018, 12 February 2019).

### 2.2. Data Collection and Outcome Measures

Data were assessed for baseline (W0), after 2 (W2), 6 (W6), 10 (W10), 24 (W24), 39 (W39), and 52 (W52) weeks (W). Due to the retrospective nature of the study, data were not available for each time point. These visit dates were chosen as they were regular follow-up visits performed at the Austrian outpatient clinics participating in the study. Baseline characteristics included demographic variables such as age, sex, AD subtype (intrinsic or extrinsic), and previous AD treatments (Table 1) [14]. Intrinsic AD is characterized by a total Immunglobulin E (IgE) level of less than 150 kU/L and no atopic comorbidities, whereas the extrinsic subtype is defined by total IgE levels above 150 kU/L and concomitant allergic diseases such as asthma or rhinoconjunctivitis [3].

For the assessment of disease severity, Investigator Global Assessment (IGA) was available for all patients. All scores were assessed by a dermatologist. Some patients were also assessed by using Eczema Area and Severity Index (EASI; *n* = 68), Scoring Atopic Dermatitis (SCORAD; *n* = 32) and Body Surface Area (BSA; *n* = 30), at the discretion of the investigator. The primary endpoint was defined as clinical response of achieving an IGA 0 or 1 and/or an IGA reduction ≥2 points at W24 compared to baseline.

EASI, SCORAD, and BSA responses, as well as IGA at other time points were defined as secondary outcomes, together with improvements in patient-oriented measures such as quality of life (Dermatology Quality of Life Index, DLQI), itch (Numerical Rating Scale itch, NRS), pain (Visual Analog Pain Scale, VAS) and sleep (sleep scale). In AD patients with additional allergic asthma, we also performed an Asthma control test (ACT) at baseline (*n* = 14) and W24 (*n* = 12) to monitor asthmatic symptoms during Dupilumab treatment at one center. An ACT score ≥20 indicates well-controlled asthma. Blood biomarkers (eosinophil counts, total IgE, lactate dehydrogenase (LDH) and eosinophilic cationic protein (ECP) levels) were available for baseline, W10, W24, and W52 in 66%, 70.2%, 68%, and 32% of patients, respectively. Total IgE level analysis was performed using the ImmunCAP(R) total IgE (Thermo Fisher Diagnostics Austria GmbH, Vienna, Austria) with an upper limit of quantification of 5000 klU/L. Finally, potential drug-related side effects were assessed for each visit.

### 2.3. Diagnostic Procedures for Rosacea-Like Folliculitis

Under Dupilumab therapy, we observed that patients developed lesions resembling rosacea in the centrofacial area and/or the thoracic region. Most patients were screened for rosacea or acne at baseline, with only a few patients already included being re-evaluated by photographs taken at baseline. All patients were asked for their medical history concerning rosacea or acne. Punch biopsies (4 mm) were obtained from affected areas of three patients with rosacea-like folliculitis and sent for histopathological evaluation. In addition, reflectance confocal microscopy (RCM; VivaScope 3000 MAVIG GmbH, Munich, Germany) was performed in all six patients (forehead and cheek bones) who developed rosacea-like folliculitis during Dupilumab therapy. RCM is a non-invasive imaging technique that allows in vivo examination of epidermal and dermal structures [15,16]. Mosaics of 10 × 10 mm^2^ were scanned on facial lesions, and the total numbers of mites per follicle and per area were counted, along with the number of follicles per area. As a control group, we also performed RCM in five gender- and age-matched AD patients who did not show this side effect during Dupilumab treatment.

### 2.4. Statistical Analyses

Metric variables were described by mean and standard deviation or by median, 25% quantile and 75% quantile. Categorical variables were described by absolute and relative numbers. Total IgE was not quantified exactly for values above 5000 units and ECP above 200 units, respectively. Values exceeding these thresholds were treated as 5000 or 200 in all calculations. The sample median and other quantiles calculated for these variables were deemed valid estimates if their value was below the threshold.

To compare mean values between two visits, a random effects model was fitted with visit as fixed effect and patient as random effect. A *t*-test and 95% confidence interval for the difference in means between the two visits were calculated from this model. Degrees of freedom were calculated using the method by Kenward and Roger [17]. Visualization of percentage of patients with an IGA-value of 0 or 1 or an IGA change of ≥2 at each time point was done by bar-plots. For the variables EASI and SCORAD, the percentage change from baseline was calculated for each time point and the mean percentage change with 95% confidence limits was plotted. For the variables itch, pain, sleep (visual analog scale) and DLQI, the mean value per visit with 95% confidence limits was plotted. All statistical analyses were performed using SPSS Statistics Version 24.0 (IBM corp., Armonk, NY, USA) and R Version 3.6. The threshold for statistical significance was set at *p* < 0.05.

## 3. Results

### 3.1. Baseline Population

The study population included 94 patients (59 male and 35 female) from five dermatological centers with a maximum follow-up time of 52 weeks. The majority of our population had extrinsic AD (89.4%), with 26 patients (27.4%) showing allergic asthma as comorbidity (Table 1). The mean IGA, EASI, SCORAD, and BSA indexes prior to treatment were 3.9 ± 0.7, 26.5 ± 12.5, 69.2 ± 11.6, and 60.2 ± 19, respectively. Patient-oriented scores such as itch, pain, sleep, and DLQI were all consistent with moderate-to-severe disease for all patients (Table 1). Serological markers at baseline showed a median total IgE of 3715 kU/L (Q1: 557.5, Q3: 5000), median ECP of 34.4 μg/L (Q1: 24.15, Q3: 61.825), and median LDH of 245.5 U/L (Q1: 205, Q3: 303.75). Patients with concomitant asthma had an initial ACT value of 18.42 ± 4.9, consistent with uncontrolled asthma (Table 1).

### 3.2. Clinical Effectiveness of Dupilumab

During the study period, all patients maintained their usual skincare regimen, including moisturizers and topical TCS/TCI for flares. Following Dupilumab initiation, 38% of patients experienced a rapid improvement of clinical disease activity achieving an IGA improvement ≥2, and 12% of patients reached an IGA score of 0 or 1 already after two weeks (Figure 1A). After 24 weeks of treatment, a substantial proportion of patients (84%) achieved an IGA improvement ≥2 and 60% of patients an IGA of 0/1 (Figure 1A).

This result was also reflected by a 78% decrease in EASI (mean 6.4 ± 6.5, *p* < 0.0001), 64% in SCORAD (24.6 ± 12.1, *p* < 0.0001), and 72% in BSA (16.9 ± 11.6, *p* < 0.0001) at W24 (Figure 1B). After 24 weeks, 31 patients achieved an EASI 50 and 25 patients an EASI 75 improvement (Appendix A). Patient-oriented scores also demonstrated a rapid and significant improvement at W24 compared to baseline (5 ± 4.6 vs. 17.3 ± 6.5 for DLQI, 1.9 ± 2.1 vs. 7.6 ± 1.7 for itch, 1.7 ± 2 vs. 7.2 ± 3 for sleep, and 1.8 ± 2.1 vs. 6.2 ± 2 for pain, *p* < 0.0001; Figure 1C). Similar values for all parameters were also observed after 52 weeks of treatment, albeit lower observational numbers were available, as not all patients had reached this time point when data were analyzed. Dupilumab treatment increased the mean ACT score in asthmatic patients leading to a significant symptom reduction after 24 weeks (22 ± 3.8 vs. 18.4 ± 4.9, *p* < 0.05). Compared to baseline, clinical effectiveness was accompanied by normalization of LDH (median 177 U/L; Q1: 155, Q3: 221), stable values in total IgE (3430.5 kU/L; Q1: 1347.5, Q3: 5000), and an increase in median ECP levels (69.5 μg/L; Q1: 38.6, Q3: 100.2) until W24. The median amount of total IgE declined after one year of therapy down to 2,114 kU/L (Q1: 1256, Q3: 2372), and ECP levels, after peaking at W24, decreased again to baseline values by W52 (46.65 μg/L; Q1: 29.1, Q3: 58.1)).

### 3.3. Safety

The administration of Dupilumab was well tolerated, with generally only mild side effects. A common adverse event was non-infectious conjunctivitis or blepharitis in 38 patients (40%), which was diagnosed by either the investigator or a consulting ophthalmologist. For most patients, eye symptoms were mild, and treatment with artificial tears was sufficient. For 25% of the patients glucocorticoid- or cyclosporine-containing eye drops were used for disease-control, and three patient terminated treatment due to severity of this side effect. Surprisingly, a total of six patients (6.4%) developed new-onset rosacea-like folliculitis between W3 and W36 (Figure 2), characterized by episodic flares with aggravations a few days after Dupilumab administrations. Clinical symptoms included erythema, flushes, and papulopustules, as well as a burning sensation of the skin in the centrofacial area (Figure 2). Two patients developed rosacea-like folliculitis on the thoracic region besides the lesions on the facial region. The involvement of the thoracic region is an atypical localization of rosacea [18]. Of note, none of these patients had a medical or family history of acne or rosacea. Only two of these patients, who developed rosacea-like folliculitis, were using concomitant topical calcineurin inhibitors. Three patients agreed to skin biopsies in the face and/or thoracic region, revealing a lymphocytic inflammatory infiltrate with neutrophilic granulocytes in a perifollicular distribution pattern, as well as within hair follicles, consistent with the diagnosis of folliculitis and perifolliculitis (Figure 2). Due to the fact that rosacea is often associated with Demodex mites, we performed reflectance confocal microscopy (RCM) on facial lesions in all six patients to quantify Demodex colonization. Demodicosis in rosacea patients is usually reflected by higher numbers of mites per hair follicle (>5) as compared to healthy individuals [16,19,20]. Indeed, rosacea-affected patients demonstrated an increased number of mites per follicle per 100 mm^2^ area (1.1; range 0.6–1.7) compared to Dupilumab-treated AD controls without rosacea-like symptoms (0.47; range 0–1.9) (Figure 3). A representative picture of an RCM image showing mites is displayed in Figure 2, as revealed by 3–5 round dots in groups per hair follicle. Noteworthy, all patients with rosacea-like lesions also suffered from blepharitis and/or conjunctivitis, with various intensity. Other side effects included limited herpes simplex infections in 14 patients (14.8%) and diffuse non-scarring hair loss of the scalp in five patients (5.3%). There were 62 patients (66%) that developed relative eosinophilia (>4% of eosinophil counts) during Dupilumab treatment. No injection site reactions were reported.

### 3.4. Discontinuation of Dupilumab

Six patients discontinued Dupilumab treatment. Three patients stopped Dupilumab therapy due to severe conjunctivitis and blepharitis after 10 (two patients) and 36 weeks. Two patients decided to discontinue due to AD flares and inadequate response to treatment after 18 weeks. One patient was lost to follow up after 30 weeks without further notice.

## 4. Discussion

In this retrospective study, we present real-life data from 94 AD patients treated with Dupilumab in five dermatological centers in Austria. In contrast to reports from other investigators, none of our patients was using concomitant systemic immunosuppressive therapies at Dupilumab initiation [21,22,23].

Overall, Dupilumab administration resulted in a rapid and significant reduction of clinical disease activity and patient-oriented disease measures. We chose IGA as primary endpoint because it is a standardized, easy to perform, and frequently used assessment tool. Moreover, health insurance companies in Austria demand IGA as a disease-monitoring parameter before Dupilumab initiation, with follow-up measures every six months. The percentage of patients in our cohort reaching an IGA of 0/1 after 10 weeks (53%) and 24 weeks (60%) was higher than reported for the primary endpoint (16 weeks) in SOLO1/2 (Study of Dupilumab Monotherapy Administered to Adult Patients With Moderate-to-Severe Atopic Dermatitis) trials (38%), [11] but mean changes in EASI (−78%) and SCORAD (−65%) in our patients at W24 were comparable with 16-week data from large clinical trials CHRONOS (Long-term management of moderate-to-severe atopic dermatitis with dupilumab and concomitant topical corticosteroids): EASI −76.6% and SCORAD −62.1%; CAFÉ (Dupilumab with concomitant topical corticosteroid treatment in adults with atopic dermatitis with an inadequate response or intolerance to ciclosporin A or when this treatment is medically inadvisable: a placebo-controlled, randomized phase III clinical trial): EASI −79.8% and SCORAD −62.4% and recent daily-practice observations from other European countries [9,10,21,22,23]. Already two weeks after treatment initiation, Dupilumab increased our patients’ overall quality of life by lowering intensity of pruritus as well as pain and restoring sleep quality, which is in accordance with results from other real-life experiences [21,22]. Along with disease amelioration we also observed changes in serological markers. LDH, known as being associated with AD severity, normalized after 24 weeks [24]. We also noticed a Dupilumab-dependent decrease of IgE levels later during the study, consistent with previous reports, and in line with the concept that IL-4 and IL-13 play a pivotal role in IgE synthesis by B-cells [25,26]. Whether this effect of Dupilumab will eventually lead to a reduction in clinical symptoms of seasonal and nutritive allergies has yet to be determined [27,28].

With regard to concomitant asthma, we observed that, in line with data from clinical phase 3 trials in asthma, administration of Dupilumab led to an improvement of pulmonary symptoms [29].

In our cohort, the most frequent side effect was blood eosinophilia (65.3%) accompanied by a peak in ECP levels at W24, which normalized by the end of our observational period, an effect that needs further investigation. We observed higher clinical response rates than reported before in clinical phase 3 trials (14%), but comparable results to an analysis recently performed in a French real-life cohort (56.5%) [9,21]. Impaired tissue homing might be the cause for blood eosinophilia, as migration of eosinophils to tissues, but not their bone marrow egress or production, is dependent on IL-4/IL-13-induced eotaxins and vascular-cell adhesion molecule-1 [30]. Eye symptoms were present in 40% of our study population. Rates of conjunctivitis/blepharitis were within the range of previously published real-life investigations (from 38.2% to 62%), but much higher than in original phase 3 clinical trials (8%) [21,23,31]. The severity of ocular side effects led to Dupilumab discontinuation in three patients, who also displayed insufficient overall skin response to Dupilumab, similar to previous reports [22]. Unfortunately, we have no information about prior occurrence of allergic conjunctivitis/blepharitis in these individuals. Currently, the etiopathogenesis of Dupilumab-associated conjunctivitis is still unclear, but several hypotheses have been postulated, e.g., insufficient goblet cell function, pre-existing subclinical inflammation being unmasked by treatment, or colonization of the Meibomian glands at the lid rim with Demodex mites [32,33,34,35,36].

Due to Dupilumab-impaired Th2 responses, IL-17-mediated ocular inflammation and Demodex colonization may be facilitated and result in Meibomian gland dysfunction in AD, similar to what is seen in rosacea [35]. This hypothesis is partly supported by our observation that six patients suffering from conjunctivitis/blepharitis also showed rosacea-like folliculitis on the face and/or on the trunk. This co-occurrence of conjunctivitis/blepharitis with rosacea-like folliculitis suggests that they might have a common pathophysiological basis [34,37]. Mechanistically, rosacea has been suggested as a Demodex–triggered TH1/Th17-skewed disease with an increased infiltrate of mast cells and macrophages [34]. Consistently, we found increases in Demodex mites in hair follicles of patients with rosacea-like folliculitis compared to Dupilumab-treated patients without these symptoms, suggesting that they might be a specific trigger for this side effect, but exact mechanisms remain to be elucidated [33,38]. In this context, it would be interesting to evaluate ocular complaints in Dupilumab-treated patients paying particular attention to the presence of Demodex mites.

Taken together, clinical effectiveness of Dupilumab in AD patients in daily practice in Austria was similar to other real-life cohorts. Given the real-life character of this study, limitations included a missing placebo control group, variable outcome assessments in clinical scores (EASI, SCORAD, BSA, VAS) and incomplete data (lab values, clinical visits) at some time points. Concerning safety, we describe the occurrence of new-onset rosacea-like folliculitis as a so far unknown side effect, as well as higher rates of conjunctivitis and eosinophilia than reported before in phase 3 clinical trials. Future studies will be necessary in order to assess specific risk factors and the mechanism responsible for the development of rosacea-like folliculitis in a subset of patients following IL-4/IL-13 inhibition.

## Figures and Tables

**Figure 1 jcm-09-01241-f001:**
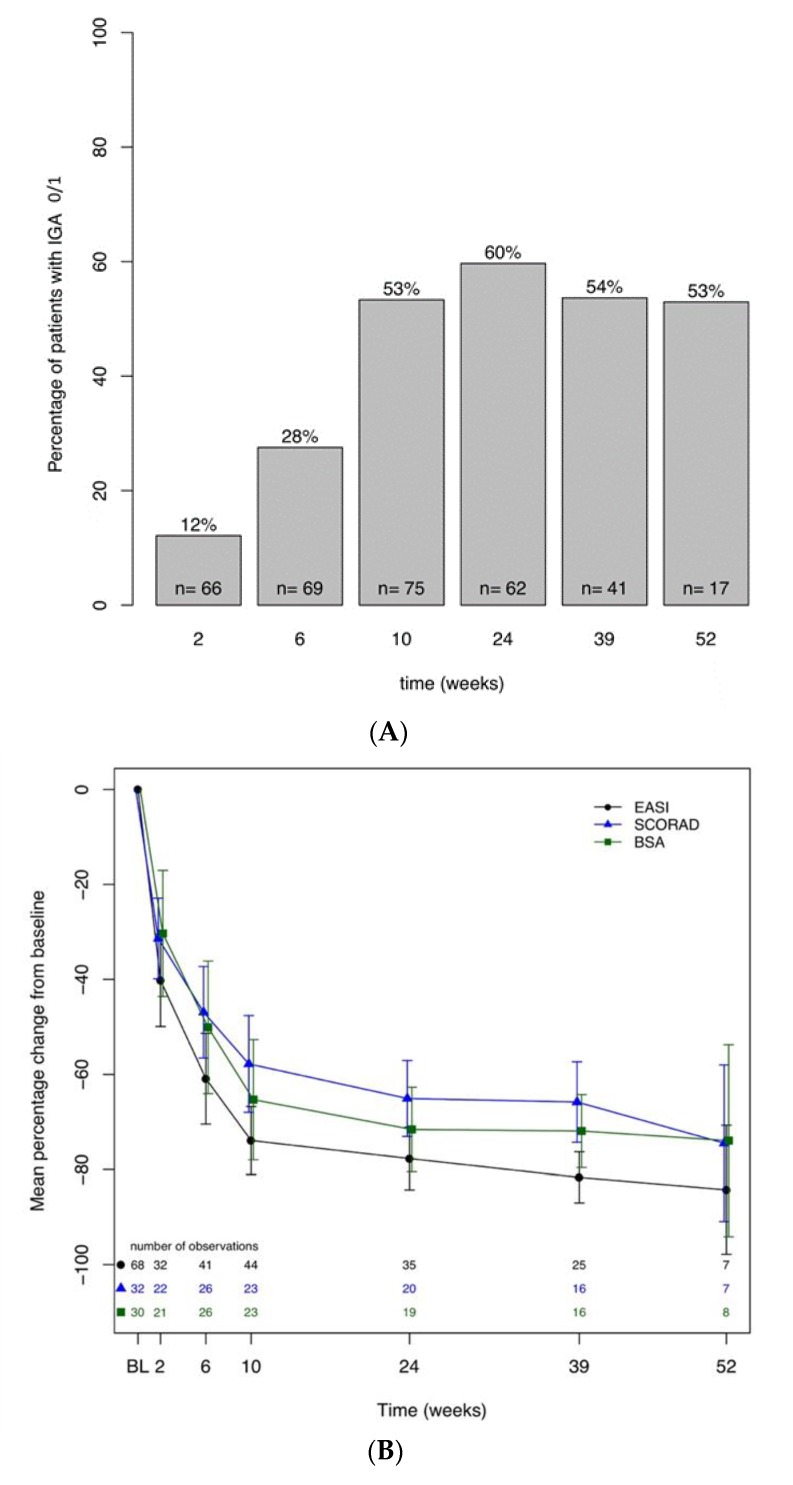
(**A**). Percentage of patients with an Investigator Global Assessment (IGA) of 0 or 1. Numbers within bars represent number of observations. (**B**). Eczema Area and Severity Index (EASI), Scoring Atopic Dermatitis (SCORAD), Body Surface Area (BSA)—Mean percentage change from baseline up to 52 weeks. (**C**). Improvement of the variables itch, sleep and pain (visual analog scale (VAS)) from baseline up to 52 weeks.

**Figure 2 jcm-09-01241-f002:**
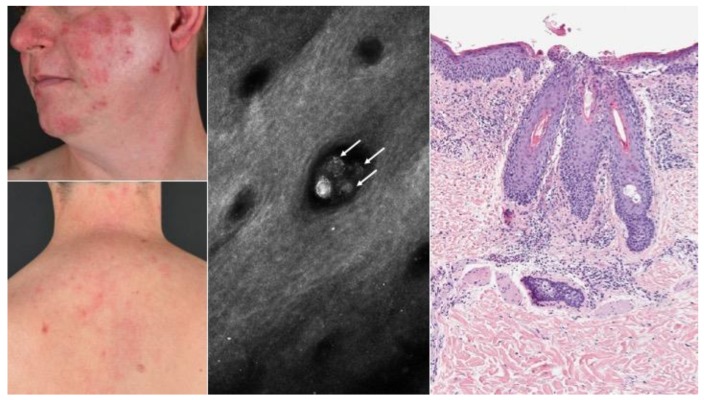
Atopic dermatitis with side effects of special interest. From left to right: Clinical picture of rosacea-like folliculitis lesions. Reflectance confocal microscopy (RCM) of Demodex mites. Mites emerge as multiple rounded structures within the hair follicles (arrows). Histopathology of a representative lesion showing typical features of folliculitis and perifolliculitis.

**Figure 3 jcm-09-01241-f003:**
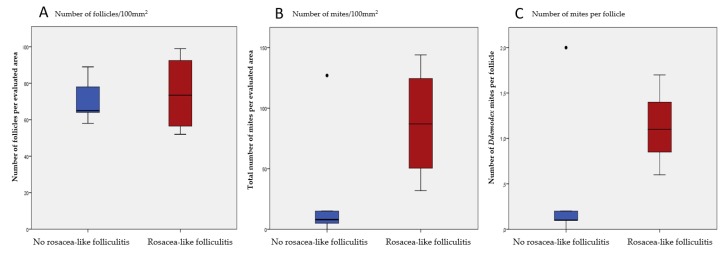
Quantification of Demodex mites and follicles in atopic dermatitis (AD) patients with (red) and without (blue) rosacea-like folliculitis. (**A**) There was no significant difference between the counted number of follicles in the rosacea-like folliculitis group and the control group. (**B**) Patients with rosacea-like folliculitis showed higher numbers of Demodex mites within an area of 100 mm^2^. (**C**) A higher number of mites per follicle was observed in patients with rosacea-like folliculitis than in controls. All results are computed as mean values including standard deviation (SD). Due to low patient numbers results are not significant but show a clear trend.

**Table 1 jcm-09-01241-t001:** Baseline characteristics (*n* = 94).

**Age**	18–87 y	
**Sex**	Male	Female
***n* (%)**	59 (63%)	35 (37%)
Atopic dermatitis (AD); *n* (%)		
AD Intrinsic (IgE levels < 150 klU/L)	11 (11.7%)	
AD Extrinsic (IgE levels > 150 klU/L)	84 (89.4%)	
Scores at Baseline		
Investigator Initiated Scores		
IGA ^a^ (0–4)	3.9 ± 0.7	
EASI ^a^ (0–72)	26.5 ± 12.5	
SCORAD ^a^ (0–103)	69.2 ± 11.6	
BSA ^a^ (0–100%)	60.2 ± 19	
Patient-Oriented Scores		
DLQI ^a^ (0–30)	17.3 ± 6.5	
Itch ^a^ (0–10)	7.6 ± 1.7	
Sleep ^a^ (0–10)	7.2 ± 3	
Pain ^a^ (0–10)	6.2 ± 2	
Biomarkers at Baseline	Value (first quartile, third quartile); sample size (*n*)	
Total IgE ^b^ klU/L	3715 (557.5, 5000); *n* = 66	
ECP ^b^ μg/L	34.4 (24.15, 61,825); *n* = 30	
LDH ^b^ U/L	245.5 (205, 303.75); *n* = 64	
Allergic Asthma; *n* (%)	26 (27.6%)	
Asthma Control Test (ACT) ^a^ (0–25)	18.4 ± 4.9	
Previous Treatments; *n* (%)		
Topical Treatments (Calcineurin Inhibitors, Corticosteroids)	94 (100%)	
Systemic Treatments	94 (100%)	
Phototherapy (NB UVB/PUVA)	61 (64.9%)	
Cyclosporine A	34 (36.2%)	
Methothrexate	15 (15.9%)	
Azathrioprine	8 (8.5%)	
Other (IVIG ^c^, Omalizumab, Rituximab)	26 (27.6%)	

^a^ Mean scores at W0 ± standard deviation; ^b^ Median score at baseline (first quartile, third quartile); ^c^ intravenous immunoglobulin. IGA: Investigator Global Assessment; EASI: Eczema Area and Severity Index; SCORAD: Scoring Atopic Dermatitis; BSA: Body Surface Area; DLQI: Dermatology Life Quality Index; NB-UVB: narrowband ultraviolet B phototherapy; PUVA: psoralen and ultraviolet A (UVA) therapy; IVIG: intravenous immunoglobulin.

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
