# Peer review of "Dupilumab for the Treatment of Atopic Dermatitis in an Austrian Cohort-Real-Life Data Shows Rosacea-Like Folliculitis"

_jcm, 2020, doi:10.3390/jcm9041241_

Round 1
Reviewer 1 Report
The manuscript is written in clear and concise language with interesting findings.
1. Please include the Introduction.
2. Was a minimum EASI score an inclusion criterium? If yes, please add.
3. Line 44: please specify ‘inefficacy of previous treatments’. Were these topical and/or oral treatments? Which ones? For how long? Was a washout period applied?
4. Line 54-55, intrinsic/extrinsic AD subtype: please add a reference, as these terms are not common knowledge among clinicians.
5. Line 53: if data were not available for all time points due to the retrospective character, why then did you chose these time points? If these are the regular daily-practice follow-up visits, please add.
6. Line 72-79: You should introduce this paragraph a bit more. The rosacea-like folliculitis is named without introduction or explanation (maybe because the Introduction is missing now). And RCM was performed in all 6 patients à what 6 patients? What did these patients have in common to be examined by RCM? Replace the number of patients who were biopsied and examined by RCM to the Result section.
7. Line 146: please specify the clinical symptoms of the rosacea-like folliculitis (papules? Pustules? Erythema? Telangiectasia? Dry Skin? Itching? Burning sensation? Other?). Which anatomical locations were involved; only face, or also other body parts?
8. 154: how do you know that the number of mites per follicle was increased? You can only say so if you have a baseline mite value of the same patient. But as this is only one RCM measurement, you should only mention the number of mites per follicle without the assumption that they are increased. Further discuss about the normal number of mites per follicle in the Discussion, instead of the Result section (lines 154-155).
9. Lines 157-158 and 212-219: very interesting finding! In the article of Thyssen et al (third paragraph; BJD 2018; Could conjunctivitis in patients with atopic dermatitis treated with dupilumab be caused by colonization with Demodex and increased interleukin-17 levels?) an additional theory about this possible relationship is given, maybe you could add it.
It would also be very interesting to further examine the ocular complaints in patients treated with dupilumab, focussing on the presence of Demodex mites.. (Discussion point).
10. Discussion: you should add a short paragraph with study limitations. Namely that outcome assessment was variable among patients (EASI, SCORAD, BSA, etc). And that lab values and visit data were not available for all patients. Moreover, topical corticosteroids were allowed during the study at flares, could this have confounded the effect of dupilumab treatment?
Author Response
Response to Reviewer 1 Comments
To begin with, we would like to thank Reviewer 1 for his valuable and important input. We changed our text according to your suggestions and feel that these changes will greatly improve the quality of our manuscript. In the following text we will address the comments point by point:
Point 1: Please include the Introduction.
Response 1: We sincerely apologize for the missing introduction. The introduction was included. (Line 33-50 marked, 33-48clean)
Point 2: Was a minimum EASI score an inclusion criterium? If yes, please add.
Response 2: This is an important question. In Austria, health insurance companies require an IGA score as a standardized disease-monitoring parameter before dupilumab initiation, with follow-up measures every 6 months. Physicians treating AD patients are required to collect IGA scores of the patients they intend to treat with dupilumab. For this reason, we chose the IGA as our primary endpoint and main inclusion criterion. The EASI score was therefore not an inclusion criterion but was collected additionally as secondary endpoint where available.
Point 3: Line 44: please specify ‘inefficacy of previous treatments’. Were these topical and/or oral treatments? Which ones? For how long? Was a washout period applied?
Response 3: The term inefficacy to previous treatments refers to inadequate disease control with one or more traditional systemic immunosuppressive agents (e.g. methothrexate, ciclosporine A). All patients also experienced a failure of topical therapy. Previous therapies are listed in table 1. For previous unsuccessful systemic therapies, a washout period was applied according to the half time of the respective medication. We included this information in the revised manuscript. (Line 60-63 marked, 58-61 clean)
Point 4: Line 54-55, intrinsic/extrinsic AD subtype: please add a reference, as these terms are not common knowledge among clinicians.
Response 4: Thank you very much for this thoughtful comment. An according explanation and reference was included in the text: “Intrinsic AD is usually characterized by a total IgE level of less than 150 kU/L and no atopic comorbidities whereas the extrinsic subtype is defined total IgE levels above 150 KU/L and concomitant allergic diseases such as asthma or rhinokonjunctivitis”. (Line 71-74 marked, 69-72 clean, Ref 3, also shown in table 1)
Point 5: Line 53: if data were not available for all time points due to the retrospective character, why then did you chose these time points? If these are the regular daily-practice follow-up visits, please add.
Response 5: We indeed chose these time points because they were regular follow-up visits at the Austrian outpatient clinics participating in the study. This statement was included in the text. (Line 68-70 marked, 66-68 clean)
Point 6: Line 72-79: You should introduce this paragraph a bit more. The rosacea-like folliculitis is named without introduction or explanation (maybe because the Introduction is missing now). And RCM was performed in all 6 patients à what 6 patients? What did these patients have in common to be examined by RCM? Replace the number of patients who were biopsied and examined by RCM to the Result section.
Response 6: We agree with the reviewer’s observation that this paragraph needs clarification. We added the following: Under dupilumab therapy we observed that patients developed lesions resembling rosacea in the centrofacial area and/or the thoracic region. Most patients were screened for rosacea or acne at baseline, only a few already included patients were re-evaluated by photographs taken at baseline. All patients were asked for their medical history concerning rosacea or acne. (Line 118-121 marked, 91-94 clean) RCM was performed in all 6 patients who developed rosacea-like folliculitis as well as in a control group of 5 dupilumab-treated AD patients who did not develop these side-effects. The control group consisted of age – and gender-matched AD patients with a similar treatment period of dupilumab. (Line 123-130 marked, 96-103 clean) Fig 3. was added for more detailed information on this topic. Importantly, we detailed the number of patients who were biopsied as well examined by RCM in the result section. (line 217-220 marked, 178-181 clean)
Point 7: Line 146: please specify the clinical symptoms of the rosacea-like folliculitis (papules? Pustules? Erythema? Telangiectasia? Dry Skin? Itching? Burning sensation? Other?). Which anatomical locations were involved; only face, or also other body parts?
Response 7: We thank the reviewer for observing that our definition of rosacea like folliculitis was not yet clearly described. We now further specified the clinical symptoms induced by rosacea-like folliculitis. Clinical symptoms included erythema, flushes, papulopustules, as well as a burning sensation of the skin in the centrofacial area and/or the thoracic region. (Line 211-214 marked, 172-175 clean Fig. 2)
Point 8: 154: how do you know that the number of mites per follicle was increased? You can only say so if you have a baseline mite value of the same patient. But as this is only one RCM measurement, you should only mention the number of mites per follicle without the assumption that they are increased. Further discuss about the normal number of mites per follicle in the Discussion, instead of the Result section (lines 154-155).
Response 8: This is a very good point. We were only able to perform the RCM analysis at one time point because these were unexpected findings during dupilumab therapy and we performed the analysis as soon as symptoms occurred. To be able to compare these results with patients without rosacea symptoms, we performed RCM in a matched control group (see also above in response 6). This was further specified in Figure 3 and its figure legend. The manuscript was also adapted accordingly. (Line 220-225 marked, 181-186 clean)
Point 9: Lines 157-158 and 212-219: very interesting finding! In the article of Thyssen et al (third paragraph; BJD 2018; Could conjunctivitis in patients with atopic dermatitis treated with dupilumab be caused by colonization with Demodex and increased interleukin-17 levels?) an additional theory about this possible relationship is given, maybe you could add it.
It would also be very interesting to further examine the ocular complaints in patients treated with dupilumab, focussing on the presence of Demodex mites.. (Discussion point).
Response 9: This is indeed an interesting suggestion. The citation as well as the additional theory was included in the discussion section as follows:”Due to dupilumab-impaired Th2 responses, IL-17-mediated ocular inflammation and Demodex colonization may be facilitated and result in Meibomian gland dysfunction in AD, similar to what is seen in rosacea.” (Line 317-319 marked, 253-255 clean, Ref 35)
In this context, it would be interesting to evaluate ocular complaints in dupilumab-treated patients paying particular attention to the presence of Demodex mites. (Line 327-328 marked, 263-264 clean)
Point 10: Discussion: you should add a short paragraph with study limitations. Namely that outcome assessment was variable among patients (EASI, SCORAD, BSA, etc). And that lab values and visit data were not available for all patients. Moreover, topical corticosteroids were allowed during the study at flares, could this have confounded the effect of dupilumab treatment?
Response 10: We agree upon the limitations of our real-life study and a paragraph with limitations of the study was added. (Line 330-332 marked, 266-268 clean). Patients with atopic dermatitis need to control acute flares of their disease by using topical corticosteroids. Due to the real-life character of this study patients were allowed topical corticosteroids/calcineurin inhibitors in addition to dupilumab in order to achieve the best possible clinical outcome. We know from phase III trials (LIBERTY-AD- CAFÉ) (Ref 10), where a combination of topical corticosteroids and dupilumab/placebo treatment were allowed, that AD symptoms in the dupilumab/corticosteroid treatment group were still significantly improved as compared to the placebo/corticosteroid group. These results indicate that although a confounding effect might be present, the clinical improvement of AD lesions is largely due to the effectiveness of dupilumab treatment.

Reviewer 2 Report
This is an interesting paper and an interesting finding, especially as it may relate to the already reported conjunctivitis/blepharitis associated with dupilumab. I think this is a study that deserves publication, as it describes a large group of patients who were rigorously studied and is not merely a case report. Please see below for areas that should be improved/discussed prior to approving this paper for publication.
1. There is no introduction. This should be written.
2. It is unclear exactly how the patients were enrolled. This is a retrospective study. Did the study group include all patients seen over a certain time period at these 5 hospitals? Were any patients excluded from the analysis?
2.3 Was Reflectance confocal performed only on facial lesions or on lesions on both the face and trunk in all patients. Though the methods says that RCM was also performed on 5 control patients, the number of mites seen in these patients is not mentioned in the paper. Please include this information.
3.1 Was rosacea-like folliculitis screened for or evaluated at during baseline or screening exam? It is noted that no patients had a history of folliculitis, but it is probably worthwhile to mention whetehr there was any evaluation or screening for rosacea like lesions at baseline or not.
It should be noted that rosacea-like folliculitis and demodicosis have been reported in patients using topical calcineurin inhibitors. As topical calcineurin inhibitors were allowed during the study, is there any information available as to whether patients with this finding had used topical calcineurin inhibitors?
It should be noted that it is unusual/atypical for rosacea to be found on the trunk. This should be noted/remarked upon.
Author Response
Response to Reviewer 2 Comments
Primarily, we would like to thank Reviewer 2 for the valuable input. We feel that the quality of our manuscript will be clearly improved after addressing your comments.
- There is no introduction. This should be written.
Response 1: We would like to apologize for the missing introduction. The introduction was now included. (Line 33-50 marked, 33-48 clean)
- It is unclear exactly how the patients were enrolled. This is a retrospective study. Did the study group include all patients seen over a certain time period at these 5 hospitals? Were any patients excluded from the analysis?
Response 2: All patients with moderate to severe atopic dermatitis who started dupilumab treatment between February 2018 and March 2019 from 5 dermatological hospitals in Austria were included in this retrospective analysis. Inclusion criteria were: age >18years, diagnosed with moderate-to-severe AD according to the revised Hanifin and Rajka criteria, IGA 3 or 4 (4 point scale) and inefficacy to previous systemic and topical treatments (methotrexate, ciclosporin A, phototherapy,..). In Austria only patients with moderate-to-severe AD who had previous unsuccessful treatments with various immunosuppressives are eligible to dupilumab treatment. Consequently, patients who did not receive dupilumab were excluded from the analysis as they did not fulfill our inclusion criteria. (Line 53-63 marked, 51-61. clean)
- Was Reflectance confocal performed only on facial lesions or on lesions on both the face and trunk in all patients. Though the methods says that RCM was also performed on 5 control patients, the number of mites seen in these patients is not mentioned in the paper. Please include this information.
Response 2.3: We thank the reviewer for this important observation. We improved this paragraph as our explanation of reflectance confocal microscopy was yet not clearly described in the methods. Reflectance confocal microscopy was performed on facial lesions on forehead and cheek bones in 6 patients with rosacea-like folliculitis as well as in a control group of 5 dupilumab-treated AD patients who did not develop these side-effects. The control group consisted of age – and gender-matched AD patients with a similar treatment period of dupilumab. Only two patients with rosacea-like folliculitis showed additional lesions on the thoracic area, but for reasons of better comparability we decided to analyze lesions on cheeks and forehead in every patient. This information was included in the paper and figure 3 was added for better understanding. (Line 118-130 marked, 91-103 clean, Fig. 3)
3.1 Was rosacea-like folliculitis screened for or evaluated at during baseline or screening exam? It is noted that no patients had a history of folliculitis, but it is probably worthwhile to mention whether there was any evaluation or screening for rosacea like lesions at baseline or not.
Response 3.1: After we noticed this uncommon side effect we started to screen new patients for rosacea or acne at baseline. Patients already included in the study were re-evaluated by photographs taken at baseline and asked for medical history for rosacea like lesions or acne. (Line 119-121 marked, 92-94 clean).
It should be noted that rosacea-like folliculitis and demodicosis have been reported in patients using topical calcineurin inhibitors. As topical calcineurin inhibitors were allowed during the study, is there any information available as to whether patients with this finding had used topical calcineurin inhibitors?
Response: We thank the reviewer for this thoughtful comment and added a suitable comment in the result section. Four of these patients did not use any topical calcineurin inhibitors before or during manifestation of rosacea-like folliculitis. Two patients were using topical pimecrolimus once daily, however both of these patients have been using this therapy for many years and have never experienced similar symptoms in the past (line 215-217 marked, 176-178. clean). We agree with the reviewer that this is an interesting finding and that this co-synergistic effect should be closely monitored. A large patient group will be needed in order to clarify these findings.
It should be noted that it is unusual/atypical for rosacea to be found on the trunk. This should be noted/remarked upon.
Response: Only two patients developed rosacea-like folliculitis on the thoracic region in addition to lesions on the frontocentral area. This is indeed an atypical localization of rosacea. This information was included in the paper and a citation was added. (Line 214 marked, 175 clean, REF 18)
